# An Experimental Characterization of TORVEastro, Cable-Driven Astronaut Robot

**Francesco Samani *** and **Marco Ceccarelli ***

LARM2—Laboratory of Robot Mechatronics, University of Rome Tor Vergata, 00133 Rome, Italy
* Correspondence: francescoelio.samani@students.uniroma2.eu (F.S.); marco.ceccarelli@uniroma2.it (M.C.)

**Abstract:** TORVEastro robot design is presented with a built prototype in LARM2 (Laboratory of Robot Mechatronics) for testing and characterizing its functionality for service in space stations. Several robot astronauts are designed with bulky human-like structures that cannot be convenient for outdoor space service in monitoring and maintenance of the external structures of orbital stations. The design features of TORVEastro robot are discussed with its peculiar mechanical design with 3 arm-legs as agile service robot astronaut. A lab prototype is used to test the operation performance and the feasibility of its peculiar design. The robot weighs 1 kg, and consists of a central torso, three identical three-degree of freedom (DoF) arm–legs and one vision system. Test results are reported to discuss the operation efficiency in terms of motion characteristics and power consumption during lab experiments that nevertheless show the feasibility of the robot for outdoor space applications.

**Keywords:** service robots; space robotics; experimental robotics; design; testing; astronaut robots





## 1. Introduction

In space, instead of astronauts, it is advisable to use service robots instead of humans and, therefore, such space service robots are increasingly playing a strategic role in in-orbit assistance operations [1]. For more than twenty years, great attention has been paid to service robots, in order to develop new robotic systems for applications as pointed out for example in [2]. Typical service robots are already developed for medical care, space exploration, demining operation, surveillance, entertainment, museum guide. Another future target is related to the increasing exploration of space because there is a growth of space debris and defunct spacecraft left in space which will affect the current and future space missions [3]. To avoid the costly and risky tasks manually handled by humans, space service robots can automatically perform tasks and services of catching useless debris in space [4]. In some cases, robot designs have come available on the market and considerable literature about service robot is published not only on technical problems [5].

According to the International Federation of Robotics (IFR), "a service robot is a robot, which operates semi or fully autonomously to perform services useful to the wellbeing of human and equipment, excluding manufacturing operations" [6]. According to ISO [7], service robots require "a degree of autonomy", which is the "ability to perform intended tasks that are based on current state and sensing, without human intervention". Service robots can have a partial autonomy that may include an interaction with the other robots or with human beings. There are also service robots that have full autonomy without those interactions. Service space robotics is considered to be one of the most promising domains for solutions On-Orbit Servicing (OOS) missions such as docking, mooring, refueling, repairing, upgrading, transporting, rescuing up to the removal of orbital debris [8]. Characteristics of a space environment are low air pressure, large thermal excursions, high solar radiation and microgravity. With these aspects in mind, human beings in space need a system helping in the activities. Consequently, it is important to implement a system capable of controlling the outdoor space environment and a rescue system [9]. The International Space Station (ISS) is a space station [10], as an artificial habitable satellite, in low

Earth orbits. The ISS is the most complex international scientific and engineering project in the history of humanity and the largest structure ever placed in space [11]. The first ISS component was launched into orbit in 1998 and the first long-term residents arrived in October 2000 [12]. The IIS is a laboratory that deals with research on microgravity and the space environment. In this laboratory, crew members have the opportunity to conduct experiments in biology, physics, astronomy, meteorology and other fields [13]. The IIS has handrails positioned on the outer surface to allow astronauts to move outside. It is very difficult and expensive to transport large or high-mass objects from Earth to a space service station [14]. In this regard, the design of the space robot must have the characteristics to be able to carry out operations in the space station but, at the same time, its mass and volume must be considered with the utmost caution [15]. Space robots play an increasingly important strategic role in assisting human beings in orbit, as service robots as pointed out for example in [16]. In April 1993, in one of the first events in the history of spacecraft, a small-sized multisensory robot was able to perform different tasks inside a spacecraft with different operating modes as reported in [17]. These activities are still carried out today, as operations that are remotely controlled or programmed by astronauts using a stereo video monitor. The robot was remotely controlled by a human operator and by computers using artificial intelligence. In these operating conditions, the robot was able to open and close the connectors (bayonet lock), it was able to assembly structures of individual parts, and it was able to capture floating objects [18].

To date, an astronaut robot in operation is Robonaut from NASA, the National Aeronautics and Space Administration as a humanoid robot. This robot was designed and developed by the Robotics Laboratory at NASA's Lyndon B. Johnson Space Center (JSC) in Houston, Texas [19]. Robonaut is able to work with astronauts and can use tools in Space in environments suitable for astronauts. The latest version of Robonaut features a robotic torso designed to assist the crew in EVA (Extra-Vehicular Activity missions) and is able to contain the tools that can be used by the crew. This robot has the ability to grasp and operate a large range of tools. With regard to the evaluation of the grasp quality measurements have been studied in relation on wrench spaces considering the non-uniformity of the wrench space in relation of the dimensions of force and torque [20]. In any case, Robonaut 2 does not have adequate protection to resist outside the space station and improvements are needed in order to allow it to move outside around the station. At NASA/JSC, the EVA Robotic Assistant (ERA) project developed a test-bed for robots in order to explore and understand the problems of astronaut–robot interactions. Together with JSC's Advanced Spacesuit Lab, the ERA team investigated robotic capabilities and tested them with space-functional test subjects in planetary surface-like situations [21].

Another astronaut service robot is Rollin 'Justin built by the ESA (European Space Agency) [22]. The robot's hands consist of four fingers. The mobile base allows the robot to operate autonomously over a long range. This space robot has a system consisting of a light arm. Unstructured, variable and dynamic environments require the space robots to act independently and without human support. Its multiple degrees of freedom allow Rollin 'Justin to pursue multiple objectives at the same time, while respecting a hierarchy of tasks. Experiments has been followed to perform intelligent robotic coworkers, in particular, Rollin 'Justin was used as controlled by the astronauts. The system has been tested by AAAI (Association for the Advancement of Artificial Intelligence), which controls Rollin 'Justin based in Munich, Germany, from Honolulu, Hawaii [23]. Another robot of particular importance in the field of space robotics is Kirobo, the first Japanese astronaut robot that is part of the JAXA (Japan Aerospace eXploration Agency) [24]. Kirobo was developed by the University of Tokyo in collaboration with the International Space Station teams. Kirobo arrived on the ISS on 10 August 2013 on JAXA Kounotori 4's H-II transfer vehicle. Kirobo has a height of 330.2 mm, a width of 177.8 mm and a depth of approximately 152 mm. The mass is about 0.9 kg, is able to speak Japanese. The robot's capabilities also include speech recognition, natural language processing, speech synthesis and telecommunications. The implementation of facial recognition of video recording was used in Kirobo [25].

The European Space Agency (ESA) and the Italian Space Agency (ASI) have reached an agreement to develop a space robotic system called Jerico, this system will be installed in the SPEKTR module of the MIR station. Jerico is a robot that has seven axes with a sensorized end-effector [26].

Currently, many strategic technologies have been developed for space robot technology. Rotex is a project for the realization of space automation to robotics. Rotex makes use of multisensory clamp technology, uses local sensory feedback control concepts in which a powerful 3D graphic simulation with delay compensation (predictive simulation) has been implemented in the telerobotic ground station [27]. The Mobile Servicing System (MSS) is a robotic system developed for the International Space Station. The system plays a key role in the construction and maintenance of the space station by moving equipment and structures around the station, assisting astronauts with Extra-Vehicle Activities (EVA) and performing other operations outside the station. The system consists of Canadarm2 which was successfully installed in-orbit by the Canadian Space Agency in April 2001, and performed its first task of assembling the Space Station during STS-104 in July 2001 [28].

The microgravity facilitates movement but makes the stability of mechanisms vulnerable even to small vibrations. Furthermore, space radiations can be dangerous for the actuators and for many other components: controllers; driver; sensors; electronic devices as pointed out in [29]. In space, there are strong changes in temperature and thermal gradients. Furthermore, in orbital stations, the energy source is very limited since energy transport is expensive and therefore the power consumption must be managed very carefully [30]. It is necessary to consider the difficulty in carrying objects: a robot should be as light and small as possible, as pointed out in [31]. Considering that space operations by astronauts are both dangerous and expensive, service robots are being used more and more often either as assistants or as substitutes in order to reduce risk and cost. Space robots can install devices while maintaining the space station and performing experiments in space [32].

Experimental characterization of TORVEastro is presented in this article. TORVEastro is a service space robot with multiple functionalities. In particular, the use of TORVEastro will be useful to repair mechanical parts in ISS. The application of a service robot in a space orbital station needs to consider the spatial characteristics as outlined in [33]. The paper reports result to analyzes a CAD modeling and a performance evaluation for design feasibility. Finite Element Analysis (FEA) is a useful numerical technique that has been used in this paper for modeling and simulating various thicknesses of link design to achieve the best compromise in terms of weight and resistance. A feasibility study is discussed through performance evaluation using kinematics and dynamic simulation results as reported in [34]. A TORVEastro robot prototype was first designed and then built in LARM2, in Rome. Three arm–legs are used both for locomotion and grasping. The robot design with three arm–legs has been conceived as inspired by a chameleon structure to have robot limbs available both for grasping and locomotion tasks, as outlined in [19]. With the proposed structure the space TORVEastro robot can move in most of the places of a space station by using rods and handrails. In particular, assuming to have more arm–legs is not suitable because the weight and the transmission complexity. Assuming that having fewer arm–legs is not suitable for locomotion and grasping because of the reduced possibility of doing multiple tasks simultaneously. TORVEastro is designed to repair mechanical parts of the ISS. During the testing activities, the validity of the robot design was verified and the performance was characterized. The CAD model was also used for 3D printing of the components. The built prototype was tested in order to verify the operational efficiency and to evaluate the performance characteristics during the basic operational activities.

## 2. TORVEastro Design

TORVEastro space robot has a cylindrical body design with three legs, each of which is made of three links [35]. The symmetrical assembly of the arm–legs makes them interchangeable and gives the possibility of adopting a structure with multi-functional end-effectors. The conceptual design in Figure 1 shows the kinematic structure with design

and operation parameters. In particular, $\alpha_{ij}$ is the joint angle of the $i$-th joint in the $j$-th arm–leg. The parameter $w_{ij}$ is the corresponding angular velocity. The parameter $\theta_{ij}$ is the corresponding joint angular acceleration of the $i$-th joint in $j$-th arm–leg. $L_{ij}$ is the link body of TORVEastro. $L_{ij}$ vector refers to the length of the $i$-th link in the $j$-th arm–leg. $R_{ij}$ is the reaction force vector. $A_j$ is the shoulder point, $K_j$ is the elbow point and $P_j$ is the extremity point of $j$-th arm–leg. $S_{ij}$ is the servomotor for the $i$-th link of the $j$-th arm–leg.

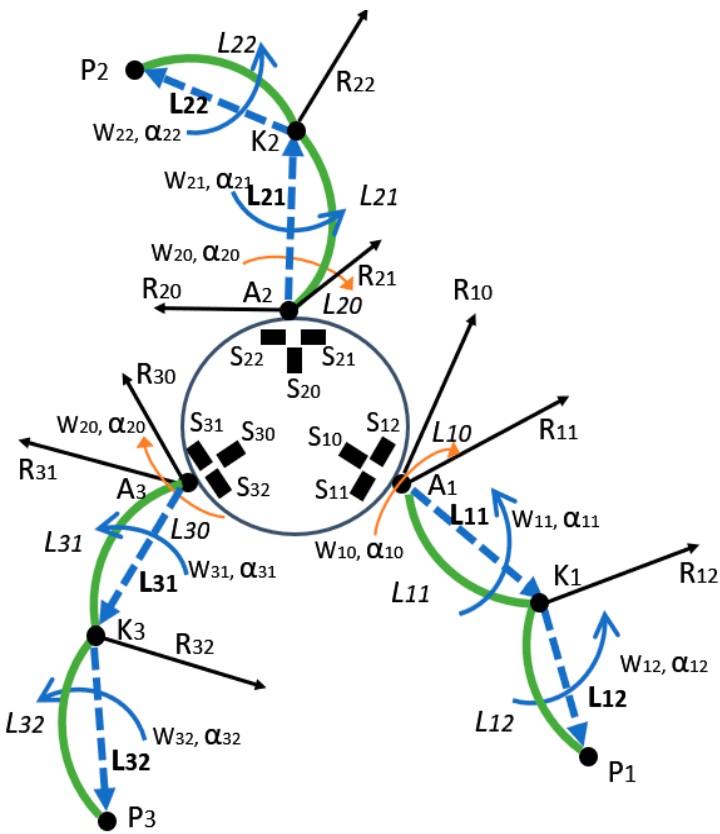

**Figure 1.** A model for the mechanical design of TORVEastro with parameters.

The central body of the robot has a cylindrical surface design and the central body of the robot has a cylindrical surface and is provided of three arm–leg limbs to perform locomotion and grasp on demand. The service robot can move along on the rods and handrails. The three arm–legs have three degreees of freedom (DoFs) with three revolute joints per arm–leg allow the rotation $\alpha_{n1}$, $\alpha_{n2}$ and $\alpha_{n3}$ ($n$ = 1,2,3), Figure 1. Three arm–legs are used both for locomotion and grasping. Arm–legs are three because one is necessary to grasp handrail, the second is necessary to grasp the second handrail to have the motion or a static posture, the third can be used for manipulation. With the proposed structure, the space TORVEastro robot can move in most of the places of a space orbital station by using rods and handrails. There are six identical links $L_{11}$, $L_{12}$, $L_{21}$, $L_{22}$, $L_{31}$, $L_{32}$ that define six correspondent vectors $L_{11}$, $L_{12}$, $L_{21}$, $L_{22}$, $L_{31}$, $L_{32}$ as related to arm–leg structures. There are other three identical links $L_{10}$, $L_{20}$ and $L_{30}$ that connect the limbs to the central body. Each link has one DoF and, in total, there are nine revolute joints that are controlled by nine servomotors ($S_{10}$, $S_{11}$, $S_{12}$, $S_{20}$, $S_{21}$, $S_{22}$, $S_{30}$, $S_{31}$, $S_{32}$). Each limb consists of three links as for example $L_{10}$, $L_{11}$ and $L_{21}$ for the first arm–leg. The first DoF gives the possibility to rotate the angle $\alpha_{10}$ and it has an angular velocity direction orthogonal to the lateral surface of the central body. The second DoF gives rotation $\alpha_{11}$ angle as shoulder rotation. The third DoF gives the possibility of an elbow rotation. Servomotors are placed inside the main body and can actuate the links by tensioned cables and limbs after transmission with spur gears. In this configuration links $L_{11}$, $L_{12}$, $L_{21}$, $L_{22}$, $L_{31}$ and $L_{32}$ are moved by tensioned cables. Cable

tension in static mode is designed of about 10 N. To have low inertia and not to expose the engines to harsh space conditions, the motors are located inside the central body and they move the links by cables. Three IMU sensors (inertial measurement unit) are used to monitor the position, velocity and acceleration of robot links. In Figure 2 a kinematic functional scheme of TORVEastro is presented when $Z_{i,j}$ axis coincides with the joint axis, i represents the number of arm–leg and j represents the number of links per arm–leg. In the revolute joints 1–2–3 of the first arm–leg being $Z_{10}$ perpendicular to $Z_{11}$, $Z_{12}$ and to the base of the central body cylinder. At the same time revolute joints 4–5–6 and 7–8–9 have a position that is a central symmetry with respect to the center of the central body with joints 1–2–3. Each link is made of a hollow cross-section of elliptical shape with a curved structure of 100 cm radius for an extension of 80°, Figure 3. The design is developed to fit with the shape of the central body in the home configuration. The internal volume of the link can be used for electrical cables wiring the components.

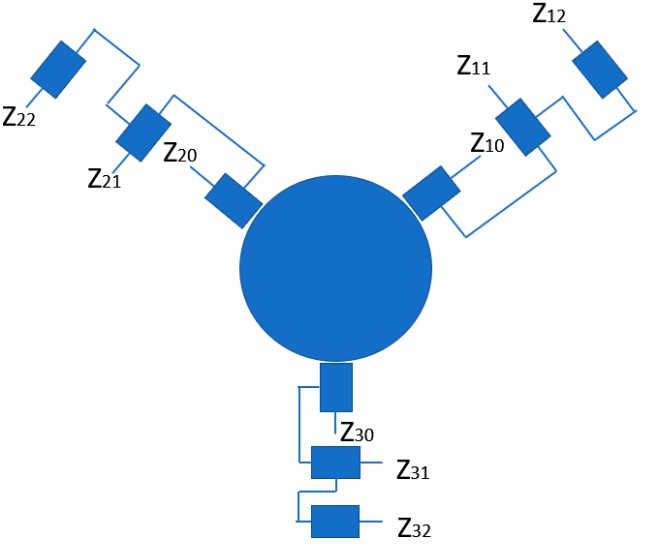

**Figure 2.** A kinematic functional scheme of the design in Figure 1.

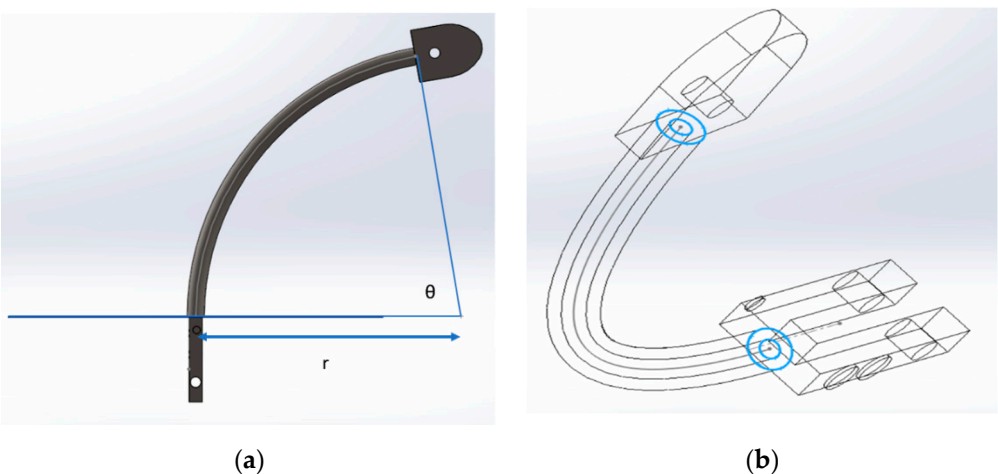

|  (a)  |  (b)  |
|---|---|

**Figure 3.** Mechanical design of a link of TORVEastro: (**a**) shape of $L_{11}$; (**b**) cross-section.

## 3. Performance Simulation

TORVEastro CAD model is designed considering the space environment. Kinematic and dynamic simulation results are used to check the feasibility of a prototype design. FEA analysis results and reaction forces show the TORVEastro robot operation peculiarities. Figure 4 summarizes the TORVEastro simulation scheme considering peculiarities in terms of

interactions with the space environment. A consideration of the environment includes also how a service robot affects or is affected by the environment, by analyzing and designing the variety of feasible conditions and situations. A dynamic simulation according to FEA analysis and kinematic analysis gives fundamental aspects for performance evaluation.

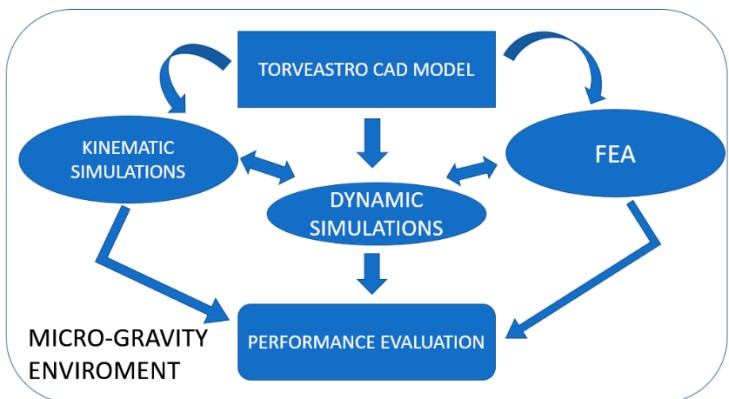

**Figure 4.** A scheme of simulation of TORVEastro.

TORVEastro robot has been simulated with a payload of 100 N. The direction of the load in the static condition is the same as the gravity acceleration. The mesh radiates from vertices to edges, from edges to faces, from faces to components, and from a component to connected components as in Figure 5a that shows link joint connection and in Figure 5b that shows joint assembly. Results of FEA analysis are reported in Figure 6 and the red arrow represents the yield strength of the material after which the deformation begins to become plastic. The stress is calculated according to Von Mises criterion, which is a scalar value of pressure (Pa) that can be computed from the Cauchy stress tensor that completely defines the state of stress of the material. The deformation in Figure 6 is highlighted by a multiplication factor of 100 times. In consideration of several computations by FEA, the final design consists of thick of 2.5 mm for $L_{11}$ and of 1.0 mm for $L_{12}$. In this way there is a proper compromise between resistance and mass (a low thick value limits the weight of the $L_{12}$ and thus the stress on the $L_{11}$) and, as shown in Figure 6, this configuration has the possibility to support the load of 100 N with suitable configuration.

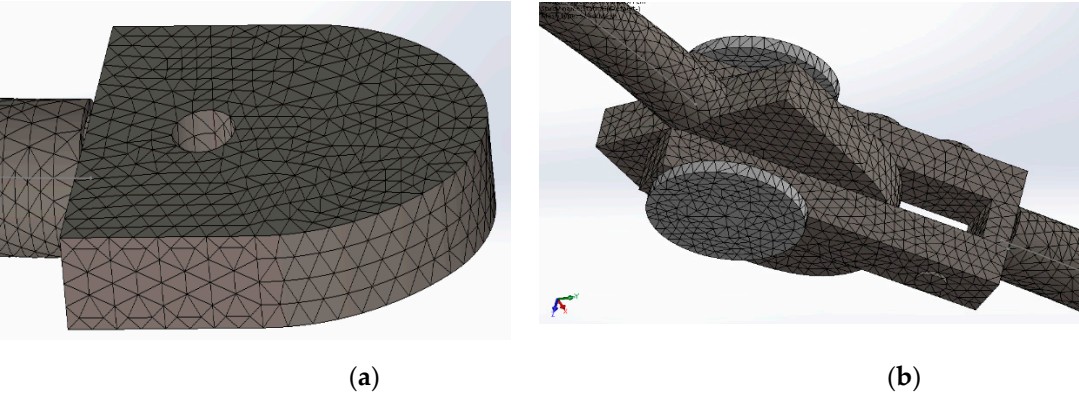

| (**a**) | (**b**) |

**Figure 5.** A mesh design for Finite Element Analysis (FEA) analysis of link design for: (**a**) link joint connection; (**b**) joint assembly.

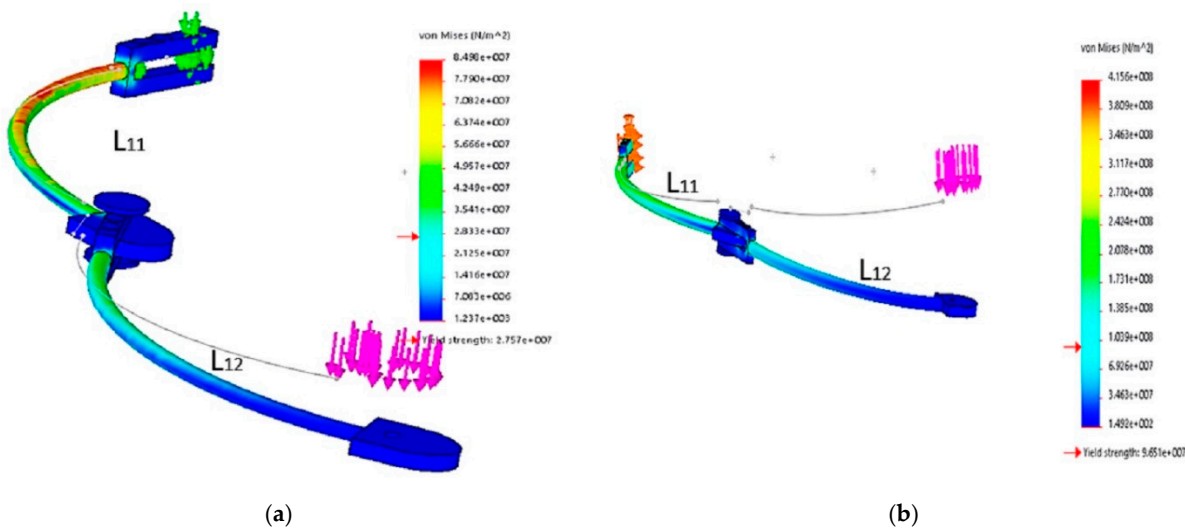

(**a**)            (**b**)

**Figure 6.** Results of FEA analysis (red line is the yield strength): (**a**) with tick of $L_{11}$ and $L_{12}$ of 2.5 mm; (**b**) with tick of $L_{11}$ is 2.5 mm and tick of $L_{12}$ of 1.0 mm.

## 4. Prototype and Testing Modes

Figure 7 shows the prototype of TORVEastro that was built at LARM2 at the University of Rome Tor Vergata. The structure of the robot was 3D printed in PLA filament (Polylactic Acid). The central body of the robot had cylindrical dimensions of 12 cm of radius and 9 cm height. The total mass of the space robot was less than 2 kg and its robot arm length was 60 cm.

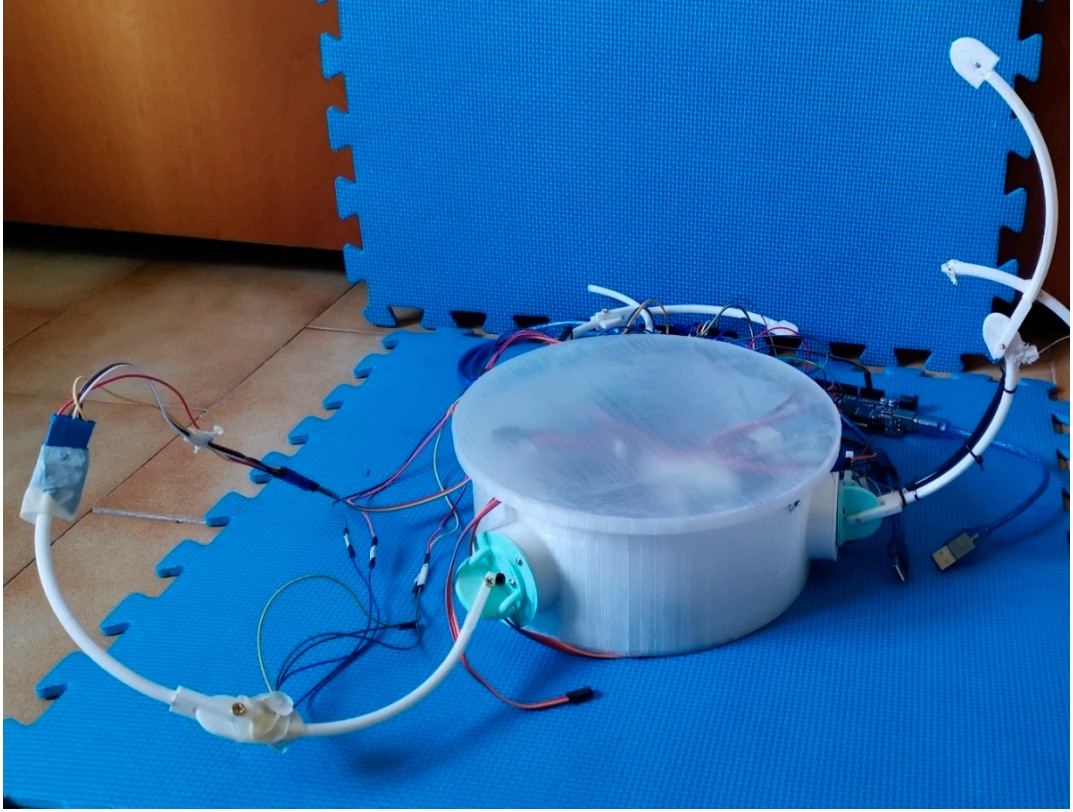

**Figure 7.** A built prototype of TORVEastro at LARM2 Rome.

The inside cross-section of the link allows the use of cables for sensors and actuators. The assembly process followed the following steps: (1) fixing the battery in the center of the central body of the robot; (2) fixing the toothed wheels; (3) fixing of links 1, 2 and 3; (4) fixing the servomotors; (5) fixing the transmissions; (6) tension testing of steel cables. Servomotors $S_{10}$, $S_{20}$ and $S_{30}$ were fixed with external teeth spur gears with module 1 and 19 teeth, Figure 8a, and external teeth spur gears transmit motion to internal teeth spur gears that have the same module 1 and 45 teeth. Actuation of links $L_{12}$ and $L_{13}$ is performed by tensioned cables (grey color), Figure 8b. Servomotors $S_{11}$, $S_{12}$, $S_{21}$, $S_{22}$, $S_{31}$ and $S_{33}$ are fixed inside the central body of the space robot. Servomotors actuate links $L_{11}$, $L_{12}$, $L_{21}$, $L_{22}$, $L_{31}$, $L_{32}$ by tensioned metal cables. The LiPo battery had 7,4 V and 6000 mAh for an operation estimated duration of 4–6 h. A webcam vision system was utilized to have visual information of the surroundings and link motion in real-time. A servomotor was mounted over the central body and webcam system is mounted on the servomotor to visualize the surrounding environment with the possibility of rotating the visual system by 360°.

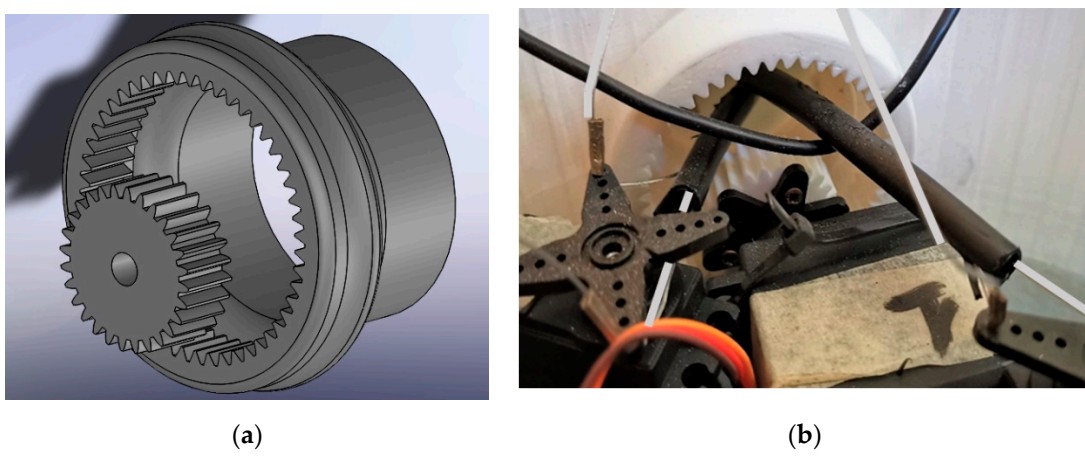

|  (a)  |  (b)  |

**Figure 8.** TORVEastro mechanical transmission in: (**a**) design solution; (**b**) a particular of tensioned cables and spur gears.

Using Arduino, it was possible to control properly the movement of the links using the testing layout in Figure 9. A LiPo battery gave power to the servomotors linked to Arduino by electric cables. A capacitor was used to reduce electrical noise and to stabilize the voltage. A capacitor with 1 µF was used to decouple servomotors of electrical network circuit from other parts. Noise by other circuit elements was shunted through a capacitor. A breadboard was used to host all the devices together. IMUs sensors (Inertial Measurement Unit) have been used and chosen according to the functional characteristics and the cost. GY-BMI 160 has a size of 13 mm × 18 mm [36]. GY-BMI 160 IMU sensors were positioned at the center of mass of each monitored link. Each IMU consisted of a three-axis gyroscope sensing acceleration with its cartesian components. The angles were calculated by integrating the velocity into the time variable, repeating the process ten times and averaging it. The current sensor was used to measure power consumption, which is very important data in space operation considering the difficulty to obtain energy in space. Servomotors MG995 [37] have 57 g of weight, like the one in Figure 10, were used to move the nine links of the robot at operating speed with these characteristics: 0.13 s/60 degrees maximum speed; maximum torque 13 kg/cm (7.2 V).

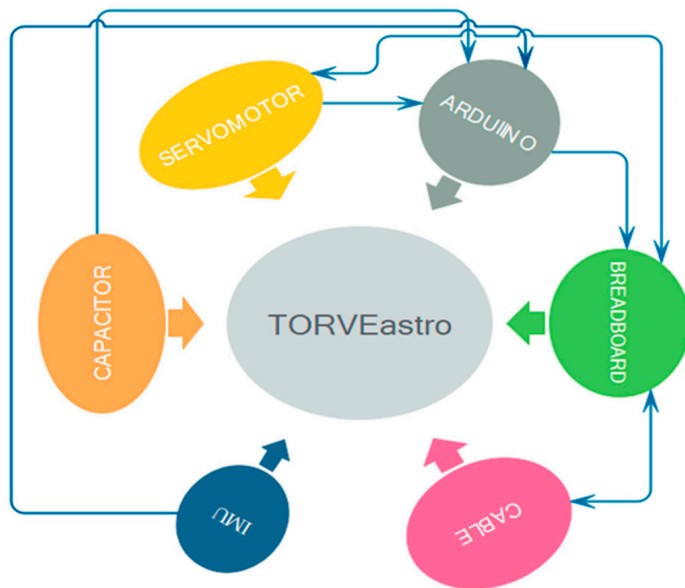

**Figure 9.** A testing layout of TORVEastro prototype at LARM2 Rome.

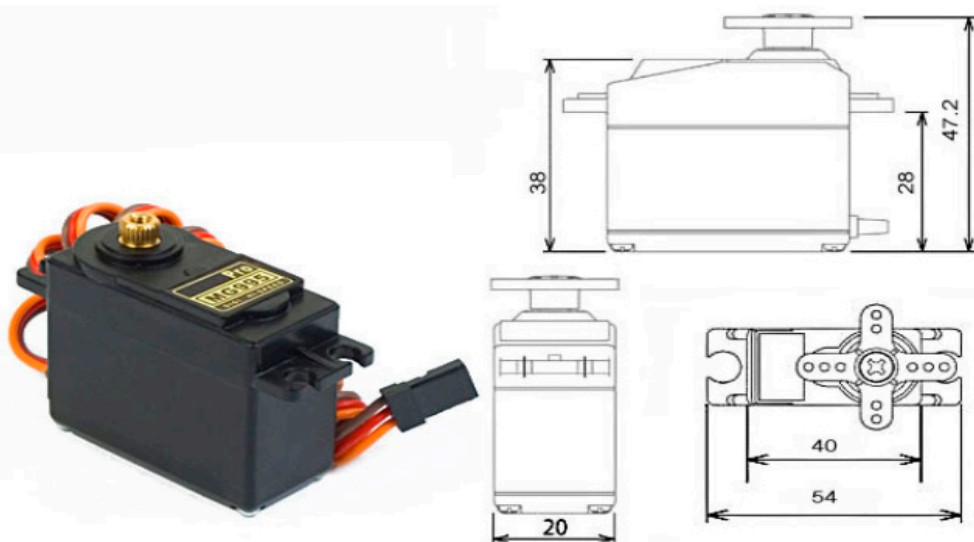

**Figure 10.** MG995 servomotor with its geometry and dimensions in mm.

Figure 11 shows an electrical scheme for an arm–leg actuation in which connections are represented for the LiPo battery, the servomotor that actuates the link, the force sensor that acquires the value of contact force, the IMU (inertial measurement unit) that measures the values of angular position, velocity and acceleration of links of the robot, the current sensor that monitors values of current and power for operational tasks.

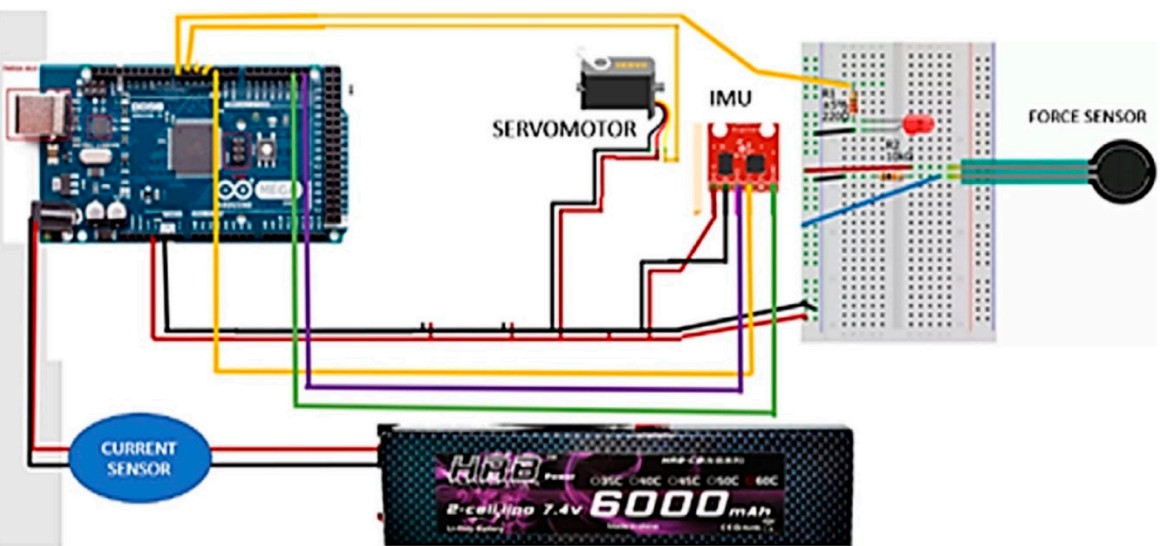

**Figure 11.** The electrical design of a single arm–leg actuation.

## 5. Experimental Results

Experimental tests of performance evaluation of the built prototype shown in Figure 11 were repeated three times to characterize the movement repeatability. The experimental results were acquired by using IMU sensors in terms of link angular position, velocity and acceleration to check TORVEastro space robot feasibility in basic operations. GY-BMI 160 IMU sensors were positioned at the center of mass of each monitored link. Links moved from start-point to end-point using different angular velocities for the three servomotors.

### 5.1. Experimental Test 1

Figure 12 shows a robot snapshot motion of the first experimental test in which only one servomotor $S_{10}$ (see Figure 1) is used at 10% of its max speed. The maximum angle range in terms of rotation is 25–35–12-degrees (respectively considering rotation around x, y and z-axes), as is reported in Figure 13. The angle values vary almost linearly resulting in a uniform speed motion. An important consideration is that in a space microgravity environment, transmission mechanisms are less stressed because they are smaller with respect to test conditions on the ground. In this test, maximal angular velocity measured 1.0 rad/s and its time evolution had comparable values in the module in the outward and return motion phases, as represented in Figure 14. Maximum angular acceleration is 1 rad/s$^2$ as shown in Figures 15 and 16 shows the angular position of link $L_{12}$ with characteristic values of 48; 12; 5 deg as maximum (respectively considering rotation around x, y and z-axis) and 5; 8; −3 deg as a minimum. Angular velocity variation is shown in Figure 17 and its value gives an indication of a smooth motion with a behavior that is characterized by a continuous motion with periodic motion following the imposed motion. Maximum angular velocity value measured 1.0 rad/s and its value shows a periodic trend and has oscillations especially when the robot arm is at the end positions. Angular acceleration variation in function of time is shown in Figure 18 with the maximum value of 1.0 rad/s$^2$ and its trend shows oscillations for vibrating motion. Power consumption with a mean value of 5 W is shown in Figure 19 and by integrating the value it is possible to estimate an operation duration of 5 h. The mean current value is 0.17 A. Experimental results of this test in terms of power consumption show no-linear variation because the servomotor continuously runs to move according to the set velocity and because the cables are always tensioned.

### 5.2. Experimental Test 2

Figures 19–22 show a snapshot and trends are shown for the angular position, velocity and acceleration in a second experimental test in which three actuators in an arm–leg work simultaneously. The input angular velocity of servomotors $S_{10}$, $S_{11}$ and $S_{12}$ is imposed at 6% of their maximum speed. The angular position of link $L_{11}$ in Figure 20 shows an angle range in terms of rotation in about 10-4-13 deg (respectively considering rotation around x, y and z-axes). The result outlines that in the three-repetition test, values are satisfactorily repeatable giving a proper link motion considering data input and data output. The maximum angular velocity value is 0.8 rad/s, Figure 21. Angular acceleration shows a strong variation as a function of time and its maximum value is 1.2 rad/s$^2$, Figure 22. The power consumption of the robot, as shown in Figure 23, is less than the sum of total power consumption of single servomotors because it does not happen that all three motors consume the maximum energy at the same time, showing a small difference between the use of one or three servomotors simultaneously. By integrating the value of power consumption, it is possible to estimate an operation duration of 7 h as related to a mean value of 9 W and by integrating the value it is possible to estimate an operation duration of 2.5 h.

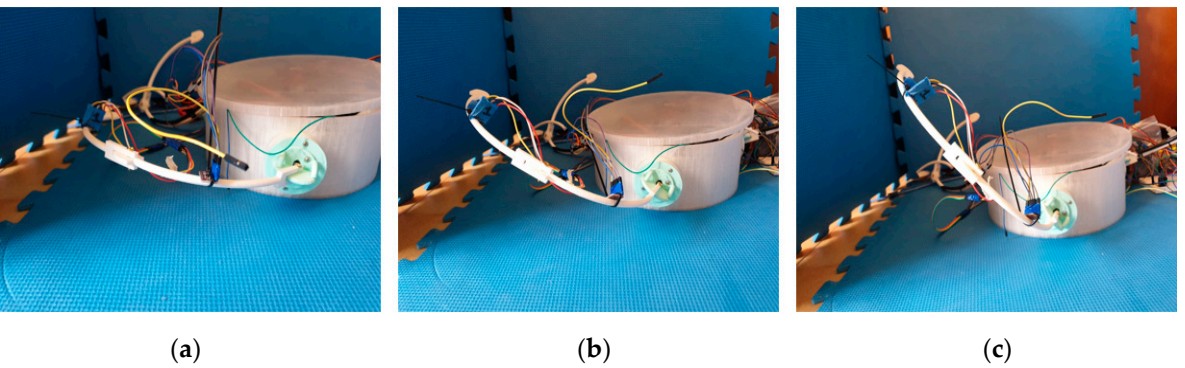

| (a) | (b) | (c) |

**Figure 12.** A snapshot during testing of the TORVEastro prototype of Figure 11 actuating $S_{10}$: in (**a**–**c**) is shown the motion of links $L_{10}$, $L_{11}$ and $L_{12}$ of the first arm–leg limb.

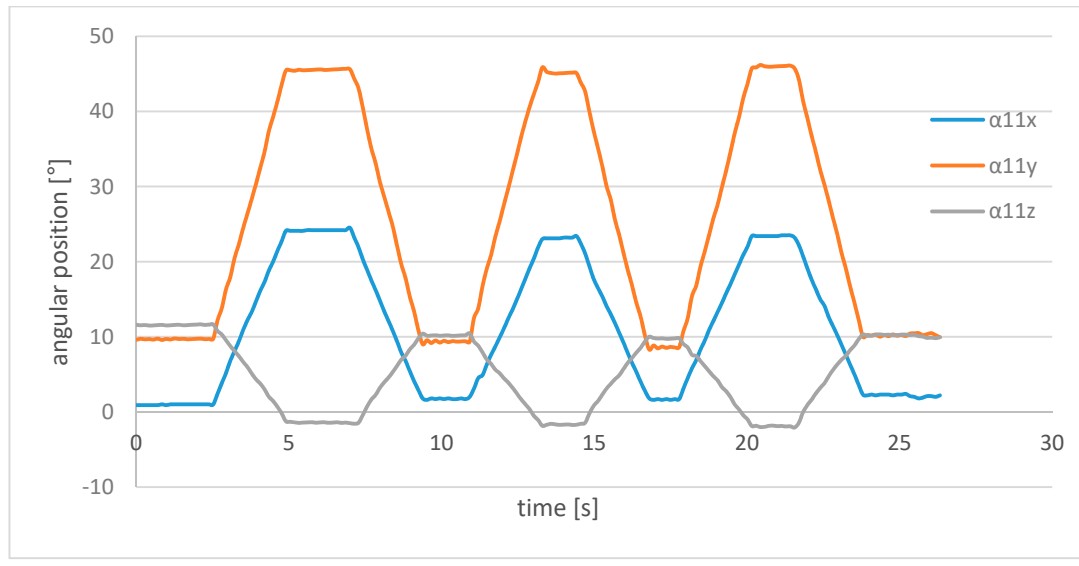

**Figure 13.** Experimental results of a test like in Figure 12 in terms of angle link $L_{11}$ of the first arm–leg limb whit only $S_{10}$ actuator moves.

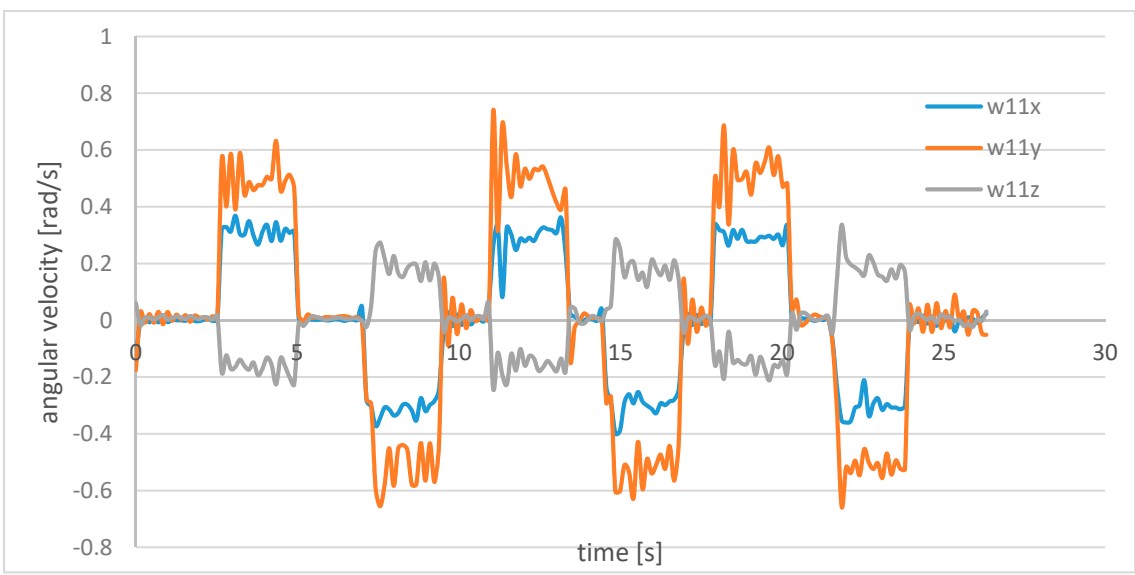

**Figure 14.** Experimental results of a test like in Figure 12 in terms of the angular velocity of arm–leg $L_{11}$ of the first arm–leg whit only $S_{10}$ actuator moves.

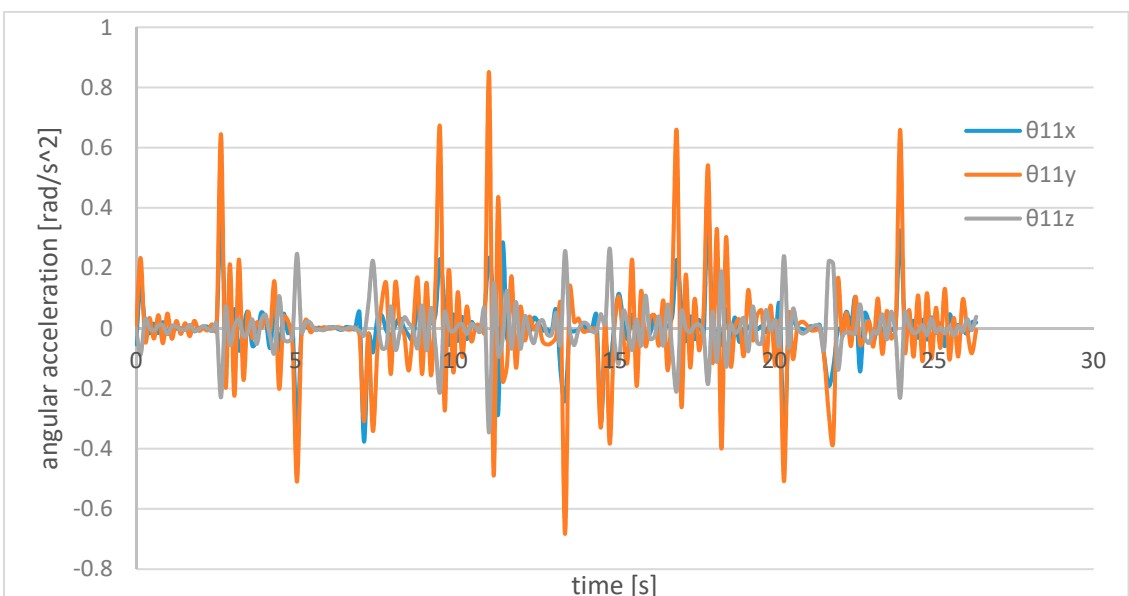

**Figure 15.** Experimental results of a test like in Figure 12 in terms of the angular acceleration of arm–leg $L_{11}$ of the first arm–leg whit only $S_{10}$ actuator moves.

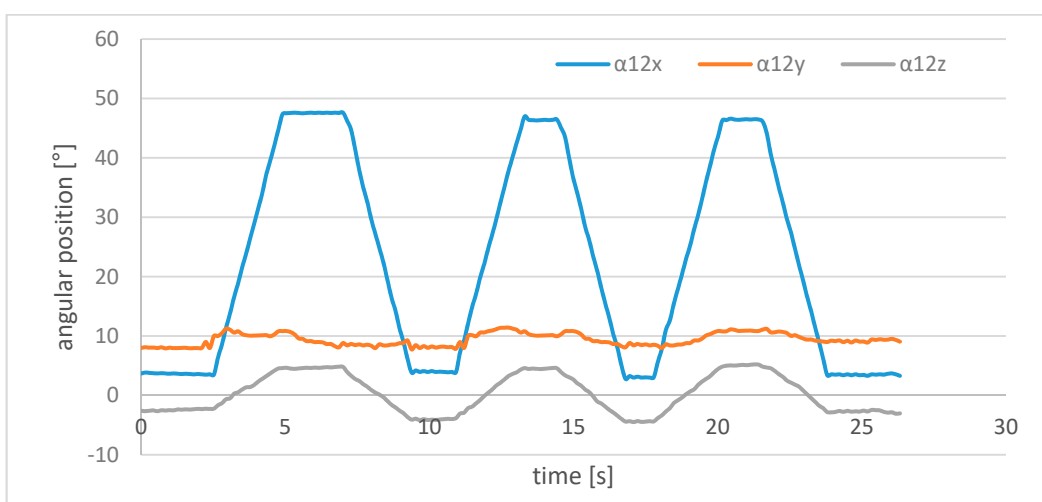

**Figure 16.** Experimental results of a test like in Figure 12 in terms of the orientation angle of arm–leg $L_{12}$ of the first arm–leg whit only $S_{10}$ actuator moves.

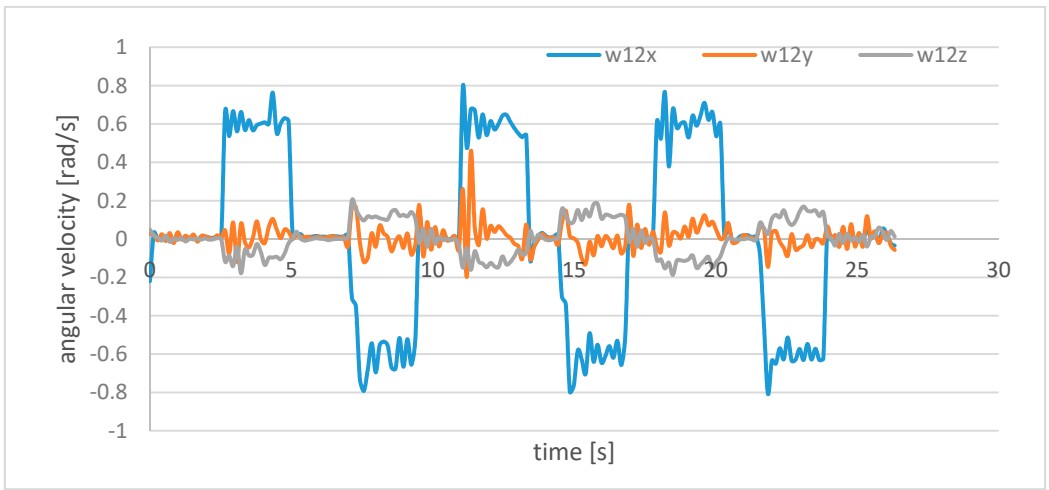

**Figure 17.** Experimental results of a test like in Figure 12 in terms of the angular velocity of arm–leg $L_{12}$ of the first arm–leg whit only $S_{10}$ actuator moves.

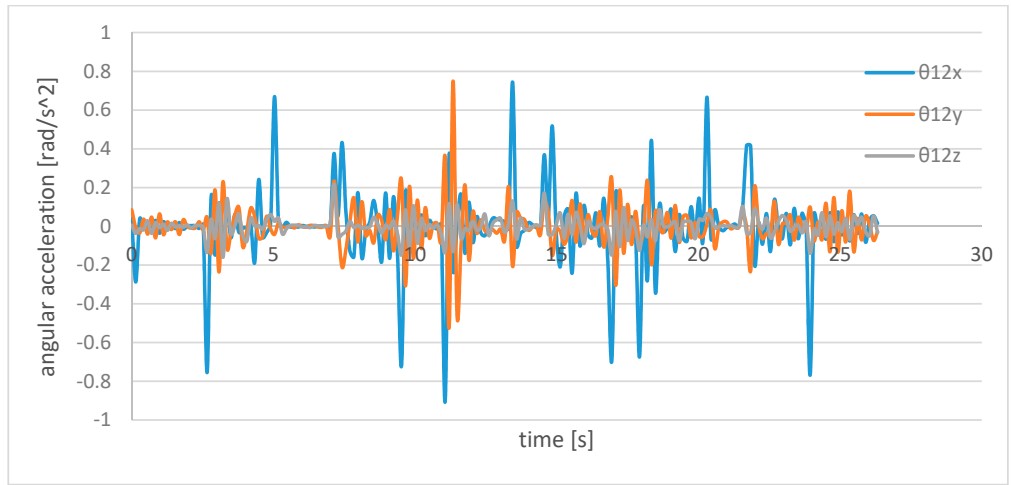

**Figure 18.** Experimental results of a test like in Figure 12 in terms of the angular acceleration of arm–leg $L_{12}$ of the first arm–leg whit only $S_{10}$ actuator moves.

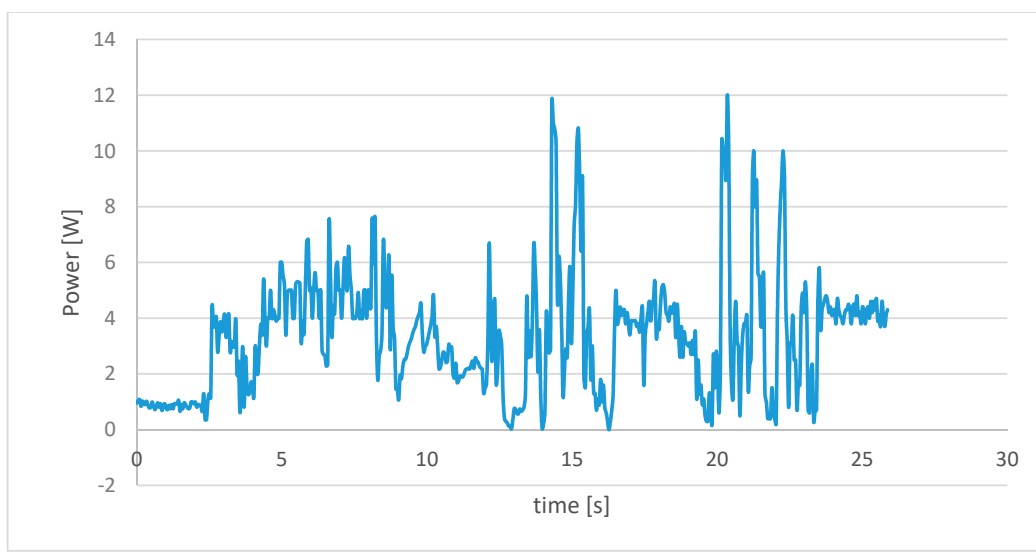

**Figure 19.** Experimental results of a test like in Figure 12 in terms of utilized power consumption for the first arm–leg whith only $S_{10}$ actuator moves.

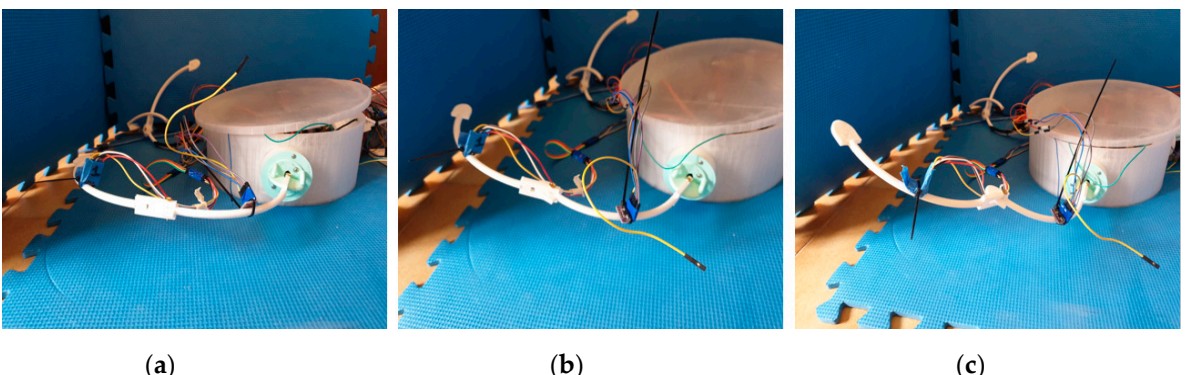

**(a)**　　　　　**(b)**　　　　　**(c)**

**Figure 20.** A snapshot during testing of TORVEastro prototype of Figure 11 actuating $S_{10}$, $S_{11}$ and $S_{12}$: in (**a**–**c**) are shown the motions of links L10, L11 and L12 of the first arm–leg, respectivley.

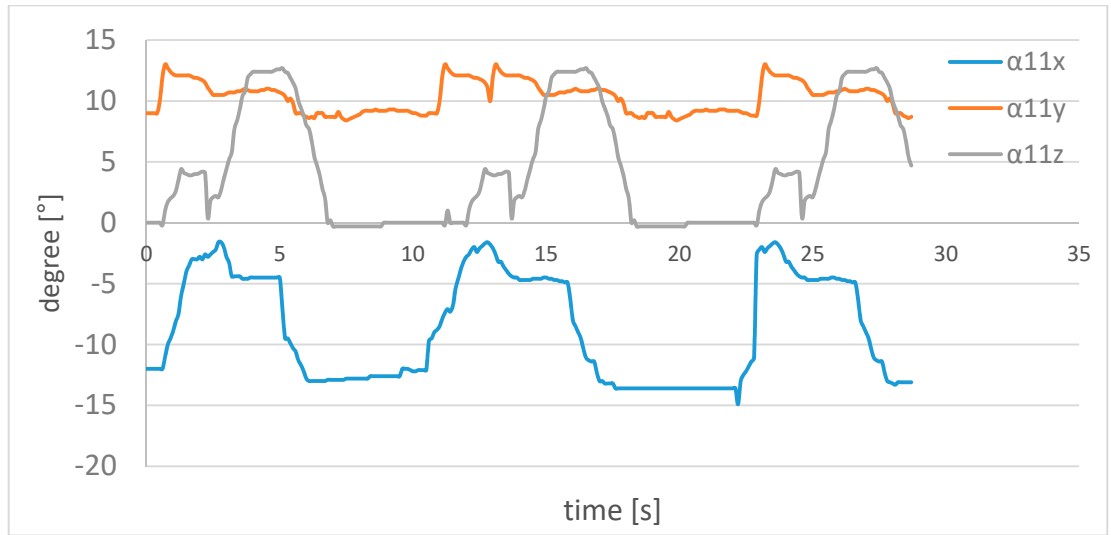

**Figure 21.** Experimental results of a test like in Figure 20 in terms of the orientation angle of arm–leg $L_{11}$ of the first arm–leg whit $S_{10}$, $S_{11}$, $S_{12}$ actuators.

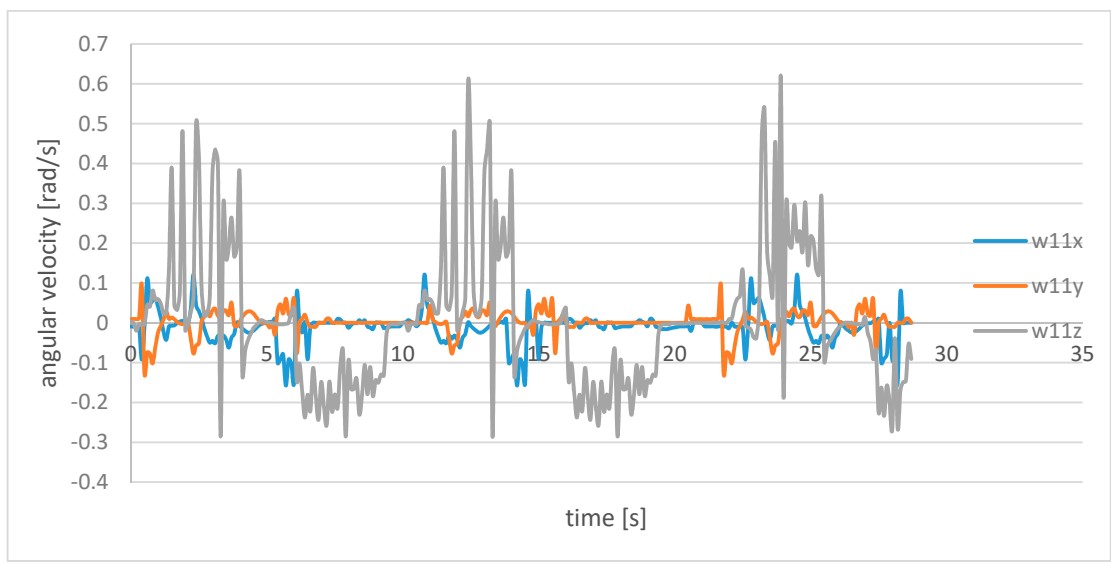

**Figure 22.** Experimental results of a test like in Figure 20 in terms of the angular velocity of arm–leg $L_{11}$ of the first arm–leg whit $S_{10}$, $S_{11}$, $S_{12}$ actuators.

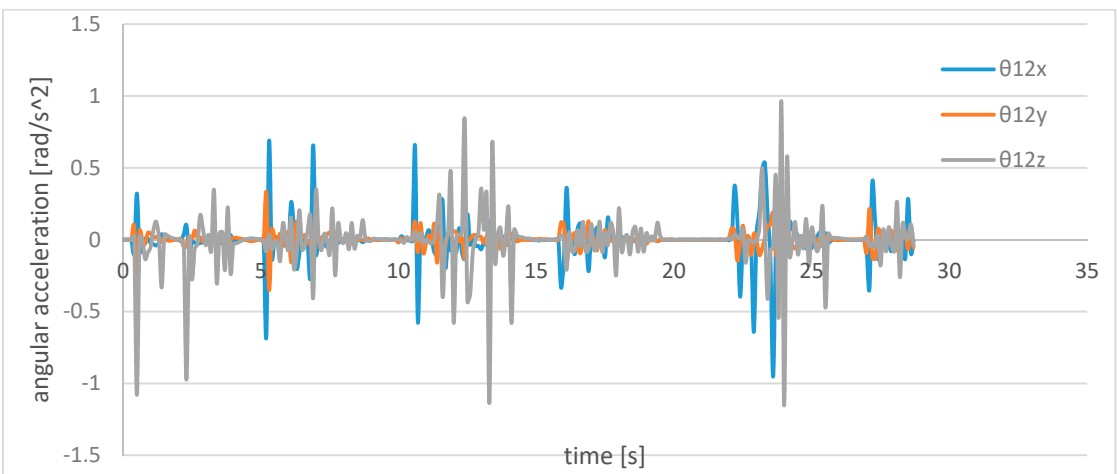

**Figure 23.** Experimental results of a test like in Figure 20 in terms of the angular acceleration of arm–leg $L_{11}$ of the first arm–leg whit $S_{10}$, $S_{11}$, $S_{12}$ actuators.

Tables 1 and 2 show a summary of input and output test results. The results show that the robot can be considered feasible for practical implementation in service space tasks since the tested TORVEastro motions and actions have been shown proper performance.

**Table 1.** A summary of experimental results of tests in Figures 12–19 with servomotor $S_{10}$ imposed motion at 10% of its maximum speed.

| Link | $L_{11}$ | | | $L_{12}$ | | | Ref. Fig. |
|---|---|---|---|---|---|---|---|
| axis | x | y | z | x | y | z | |
| $\alpha$ range [°] | 1; 25 | 10; 46 | −1; 12 | 4; 48 | 8; 11 | −4; 6 | 13,16 |
| w range [rad/s] | −0.4; 0.4 | −0.7; 0.8 | −0.2; 0.3 | −0.8; 0.8 | −0.2; 0.4 | −0.2; 0.2 | 14,17 |
| θ range [rad/s^2] | −0.4; 0.3 | −0.7; 0.8 | −0.2; 0.2 | −0.8; 0.8 | −0.6; 0.8 | −0.2; 0.2 | 15,18 |

**Table 2.** A summary of experimental results of tests in Figures 20–27, with servomotors $S_{10}$, $S_{11}$ and $S_{12}$ imposed motion at 6% of their maximum speed.

| Link | $L_{11}$ | | | $L_{12}$ | | | Ref. Fig. |
|---|---|---|---|---|---|---|---|
| axis | x | y | z | x | y | z | |
| $\alpha$ range [°] | −15; −2 | 8; 14 | 0; 14 | −5; 4 | 10; 18 | −3; 25 | 21,24 |
| w range [rad/s] | −0.2; 0.2 | −0.1; 0.1 | −0.3; 0.6 | −0.4; 0.3 | −0.3; 0.4 | −0.3; 0.8 | 22,25 |
| θ range [rad/s^2] | −0.6; 0.6 | −0.3; 0.3 | −1.1; 1 | −0.5; 0.5 | −0.4; 0.3 | −1; 1 | 23,26 |

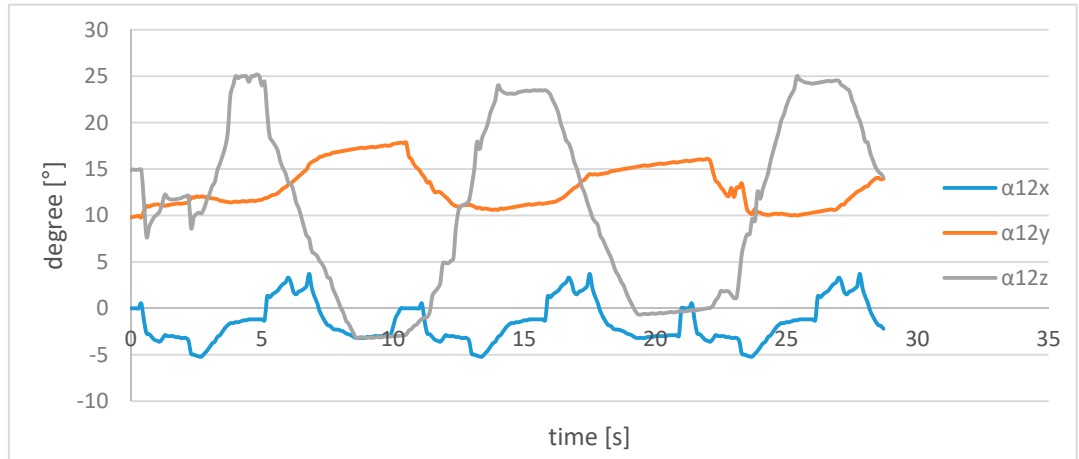

**Figure 24.** Experimental results of a test like in Figure 20 in terms of the angular position of arm–leg $L_{12}$ of the first arm–leg whit $S_{10}$, $S_{11}$, $S_{12}$ actuators.

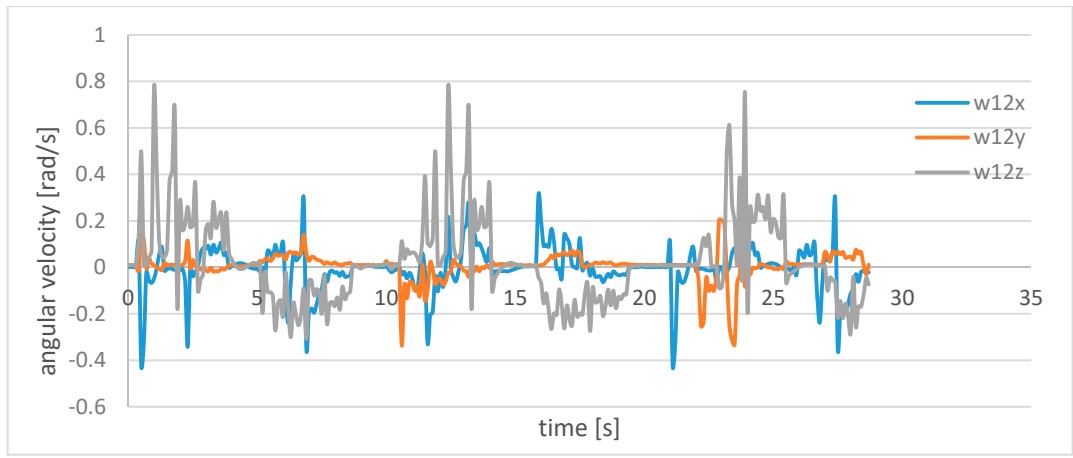

**Figure 25.** Experimental results of a test like in Figure 20 in terms of the angular velocity of arm–leg $L_{12}$ of the first arm–leg whit $S_{10}$, $S_{11}$, $S_{12}$ actuators.

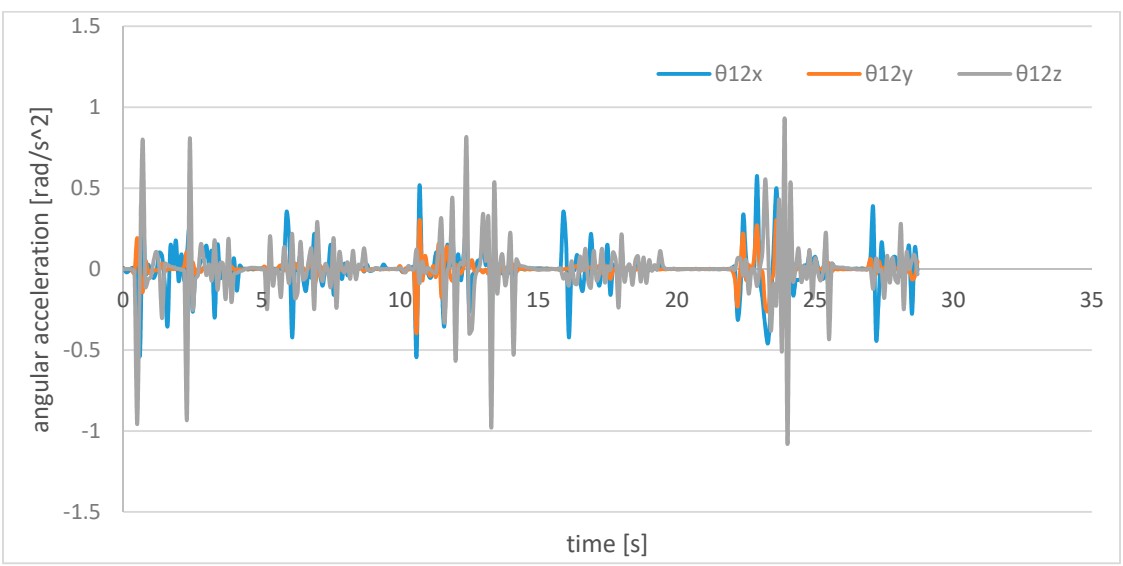

**Figure 26.** Experimental results of a test like in Figure 20 in terms of the angular acceleration of arm–leg $L_{12}$ of the first arm–leg whit $S_{10}, S_{11}, S_{12}$ actuators.

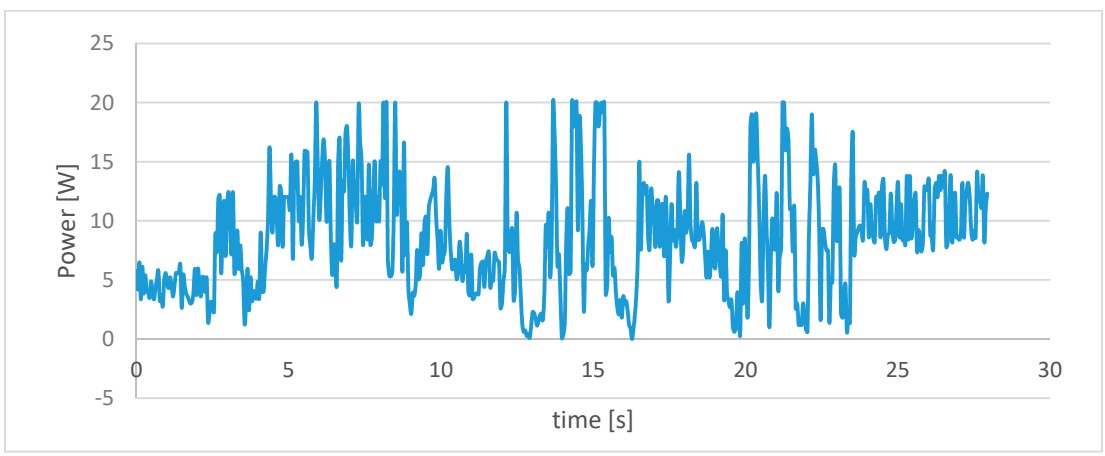

**Figure 27.** Experimental results of a test like in Figure 20 in terms of utilized power consumption for the first arm–leg whit $S_{10}, S_{11}, S_{12}$ actuators.

## 6. Conclusions

The TORVEastro robot design is presented with experimental characterization of a lab prototype looking at its basic performance. Design issues and solutions are discussed to illustrate the built prototype that has been used for performance characterization. The built prototype is designed with low-cost solutions for ground testing to check the feasibility of the arm–leg operation. Tests are reported for the basic operation of one arm–leg limb when actuated by one and three servomotors. Results of experimental tests show the suitable capability of arm–leg motion both in terms of motion performance and power consumption as feasible for an astronaut robot in monitoring and maintenance tasks in outdoor space of the orbital stations. In particular, prototype testing has shown motions of arm–leg at an angular velocity of 1 rad/s with an extremity point acceleration of 0.54 m/s$^2$ for an estimated autonomy of 1.5 h using all nine servomotors.

**Author Contributions:** Data curation, M.C.; Formal analysis, M.C.; Methodology, M.C.; Project administration, M.C.; Supervision, M.C.; Writing—original draft, F.S.; Writing—review & editing, F.S. All authors have read and agreed to the published version of the manuscript.

**Funding:** The financial support of the Italian National Project ISAF, Integrated Smart Assembly Factory is acknowledged within grant n. ARS01_01188.

**Institutional Review Board Statement:** Not applicable.

**Informed Consent Statement:** Informed consent was obtained from all subjects involved in the study.

**Acknowledgments:** Italian National Project ISAF, Integrated Smart Assembly Factory.

**Conflicts of Interest:** The authors declare no conflict of interest.

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
