# Peer review of "An Experimental Characterization of TORVEastro, Cable-Driven Astronaut Robot"

_robotics, doi:10.3390/robotics10010021_

Round 1

Reviewer 1 Report

The paper presents the study of a robot for space exploration composed of the main body and three identical 3DoF arm/legs.  The application is for sure interesting and challenging and the proposed design looks promising.

However, I have several concerns related to the proposed work. Most of them are related to the paper presentation, that could be greatly improved. 

I think that the use of the robot and the main design choices have not been clearly stated: why three legs? Are they used for what? For locomotion or grasping? 

Regarding the robot's mechanical structure, it’s not clear how the distal joints are actuated by the servomotors placed on the main body of the robot. Which type of mechanical transmission has been used?  

Regarding the paper presentation, I listed in the following my main observation. Besides them, I would recommend to the authors a careful review to improve its overall quality.

L. 105-106 —International system units are preferable L. 135 a sentence introducing TORVEastro could be useful  

Figure 1 has not a sufficient quality and clarity, some parts (e.g. the end effectors) have really low resolutions, the fonts for symbols and the stroke thickness are unnecessarily large, the overall figure could be better proportioned. Joints and links are not evident.  

L. 161-162: the same symbol is used for both the vectors and the links, it’s a bit confusing for the reader. A more formal and coherent approach should be adopted in mathematical symbols.

Overall the description of robot kinematic structure is rather confusing and should be revised. Also the terminology should be revised (e.g. rotoidal—> revolute or hinge joint)  

Figure 4: minor comment. Too many different styles for arrows have been used in this scheme, it is confusing and difficult to read. Is the color background necessary?

A simple and clean style is better in schemes and diagrams. Also, the text should be revised (cinematic—> kinematic)   L. 173-174 where are IMU sensor positioned? 

L. 198-199 Also the direction of the load should be specified.  

L. 205-206 a different symbol is used here to indicate the links.   

L. 226-227 also in this case different symbols have been used for indicating links   

Figure 8: same comment of Figure 4  

L. 271: how was the IMU used to evaluate joint angles?  

Section 5: a single and very long paragraph discusses all the experimental results, it is very difficult to follow. I would suggest to reorganise it in sub paragraphs.   

Here I further listed the lines where I found sentences that are redundant or not sufficiently clear L. 18 L. 57 L. 94 L. 138 L. 161  

By the way, I think that the proposed robot is interesting and a better presentation and the integration with some important details on robot features could solve most of my concerns. 

Author Response

REPLY TO REVIEWERS

Ms. Ref. No.:  robotics-1014938

Title: An experimental characterization of TORVEastro, cable-driven astronaut robot

Journal/Conference: Robotics,Advances in European Robotics

Reviewer 2 Report

This manuscript is application-oriented, and the paper is well organized, but some figures should be enhanced for more readable. Moreover, some details are missing, that should be added in the revised version, such as dynamics and control. Including the stability analysis on the designed robot, i.e., using Lyapunov method analyzes the stability of dynamic motion as desired. I also suggest that the authors well redraw the figures. I am willing to see the stability analysis for further consideration. If authors can provide a comparison for test the superiorty, it is better.

Q1. The quality of figues should be improved, such as, figure 1,4,8,14-27, some of which are suggested to be redrawn by python or visio.

Q2. More design details should be added in the revised version.

Q3. The dynamics of the robot designed are suggested to introduce in the revised version, the control scheme of which is essential to add as well.

Author Response

(The authors gave the same response as above.)

Reviewer 3 Report

Technical quality of the development and its presentation is not enough to attempt publication in an archived journal. The technical soundness, formatting, materials and development is very limited.   

the first filter any paper should pass is the technical level, quality of presentation, professional formatting, English proficiency; and I do not think the paper meet the standards.

Author Response

(The authors gave the same response as above.)

Round 2

Reviewer 2 Report

All my questions have been addressed.

Author Response

Thank you for your comments.